# Management of Oxaliplatin-Induced Peripheral Sensory Neuropathy

**DOI:** 10.3390/cancers12061370

**Published:** 2020-05-27

**Authors:** Guido Cavaletti, Paola Marmiroli

**Affiliations:** 1Experimental Neurology Unit, School of Medicine and Surgery and Milan Center for Neuroscience, University of Milano-Bicocca, via Cadore 48, 20900 Monza, Italy; paola.marmiroli@unimib.it; 2Department of Biotechnology and Biosciences, University of Milano-Bicocca, Piazza della Scienza 1, 20126 Milan, Italy

**Keywords:** oxaliplatin, neurotoxicity, acute, chronic, prevention, treatment, pain, neuropathy

## Abstract

Oxaliplatin-induced peripheral neurotoxicity (OIPN) is a severe and potentially permanent side effect of cancer treatment affecting the majority of oxaliplatin-treated patients, mostly with the onset of acute symptoms, but also with the establishment of a chronic sensory loss that is supposed to be due to dorsal root ganglia neuron damage. The pathogenesis of acute as well as chronic OIPN is still not completely known, and this is a limitation in the identification of effective strategies to prevent or limit their occurrence. Despite intense investigation at the preclinical and clinical levels, no treatment can be suggested for the prevention of OIPN, and only limited evidence for the efficacy of duloxetine in the treatment setting has been provided. In this review, ongoing neuroprotection clinical trials in oxaliplatin-treated patients will be analyzed with particular attention paid to the hypothesis leading to the study, to the trial strengths and weaknesses, and to the outcome measures proposed to test the efficacy of the therapeutic approach. It can be concluded that (1) prevention and treatment of OIPN still remains an important and unmet clinical need, (2) further, high-quality research is mandatory in order to achieve reliable and effective results, and (3) dose and schedule modification of OHP-based chemotherapy is currently the most effective approach to limit the severity of OIPN.

## 1. Introduction

Oxaliplatin (OHP)-induced peripheral neurotoxicity (OIPN) is a severe and potentially permanent side effect of cancer treatment [1,2]. It affects the majority of OHP-treated patients, mostly with the onset of acute symptoms, but also with the establishment of a chronic sensory loss that is supposed to be due to dorsal root ganglia (DRG) neuron damage [3,4,5].

Acute OIPN affects at least 80–90% of OHP-treated patients [6]. It consists of cold-induced paresthesias, with predominant oropharyngeal, hands, and feet distribution. These sensory symptoms ensuing hours after OHP administration are frequently associated with cramps and fasciculations, and they tend to disappear within 48–72 h in most patients. Although transitory, acute OIPN is disturbing for the patients. Moreover, it has been reported that patients with more symptoms of acute OIPN are also those who will develop more severe chronic neurotoxicity [7]. This observation is relevant since, although it does not necessarily imply a direct causal relationship, it might however be considered as evidence of higher susceptibility of some individuals to peripheral nervous system damage. The incidence of chronic OIPN is variable according to the assessment methods used to diagnose its occurrence. However, it can be considered a frequent side effect, and in the more severe cases it might markedly impair the quality of life of the affected patients. At its onset, it is characterized by numbness and tingling in hands and feet, with a distal-to-proximal extension of symptoms after increasing exposition to OHP. Once chronic OIPN progresses, sensory ataxia becomes evident manifesting with difficulty in manipulating small objects (particularly if not looking at them), standing unless base widening, and in general in all those situations where the effective balance due to proprioceptive input cannot be compensated by visual input, such as in poorly lit environments. Only anecdotally cranial, autonomic, or motor nerve impairment has been reported as a consequence of OHP administration [8].

The clinical manifestations of OHP neurotoxicity provide important clues to understand the basic mechanisms of its onset. The time course of acute OIPN clearly suggests an interaction with cellular targets able to rapidly allow the onset of symptoms, and the complete reversal of these symptoms over a few days implies a functional, rather than structural, impairment. Reversible interference with ion channels present on the DRG plasma membrane has been postulated as the mechanism at the basis of acute OIPN. In fact, OHP is able to slow the inactivation of voltage-gated Na+ channels, an effect that may be enhanced by exposure to cold [9,10,11,12,13]. Moreover, cooling can slow the kinetics in the activation of axonal slow K+ (Kv7) channels, thus modifying axonal excitability [14]. The validity of this ion channel-interference hypothesis has been confirmed by animal studies [15], and validated in small cohorts of OHP-treated patients using nerve excitability tests, a non-standard neurophysiological assessment method [10,16,17]. However, this might not be the only mechanism at the basis of acute OIPN. For instance, it has been recently reported that concentrations of OHP similar to those found in plasma of treated patients lead to an acidification of the cytosol of mouse dorsal root ganglia neurons in culture and in vivo, and this in turn is responsible for sensitization of TRPA1 channels [18]. In a subsequent study, it has been demonstrated that OHP leads to a reduction of intracellular pH by forming adducts with neuronal hemoglobin, which acts in this setting as a proton buffer and that drugs that inhibit carbonic anhydrase (an enzyme that is linked to hemoglobin in intracellular pH homeostasis), i.e., topiramate and acetazolamide, revert OHP-induced cytosolic acidification of DRG of treated animals and acute OIPN, while not affecting OHP-induced cytotoxicity on cancer cells [19].

More limited is the knowledge of the basic mechanisms of chronic OIPN. The anticancer effect of OHP is due to its capacity to bind and damage nuclear DNA. Similarly to other platinum compounds (e.g., cisplatin, carboplatin), its cytotoxicity is thought to result from inhibition of DNA synthesis following the formation of both inter- and intra-strand cross-links which prevent DNA replication and transcription, causing cell death. Based on this assumption, after the description of the first animal model of chronic OHP confirming DRG nucleolar, nuclear, and somatic size reduction with nucleolar segregation in the treated rats [20], several studies investigated at the preclinical level the effect of OHP administration on DRG neurons. However, it was subsequently demonstrated that, despite the inability of the drug to cross the blood-brain barrier, OHP effects involves also the spinal cord, causing wide dynamic range neurons hyperexcitabilty [21] and glial activation [22]. Among the pathogenetic mechanisms that have been more extensively investigated, mitotoxicty, investigated in terms of changes in mitochondrial morphology, bioenergetics, and reactive oxygen species (ROS) generation, gained a prominent position. Podratz and colleagues [23] demonstrated in vitro for the first time that cisplatin binds mitochondrial DNA in DRG neurons at the same rate as in nuclear DNA. However, unlike nuclear DNA, mitochondrial DNA does not have any DNA repair system (e.g., base excision repair, nucleotide excision repair pathways) and, therefore, platinum adducts cannot be removed causing problems in mitochondrial protein synthesis, impairing the mitochondrial respiratory chain functionality and, eventually, leading to energy failure and oxidative stress. These observations were subsequently extended and in a rat model of OIPN reported by Xiao and Bennet [24] it was shown an increase in swollen and vacuolated mitochondria in saphenous nerves of treated animals compared to controls. This pathological observation was in agreement with the results reported by Zheng and co-workers [25] showing a deficit in respiration rate followed by a decrease of ATP production in isolated sciatic nerves obtained from OHP-treated rats. The mitotoxicity hypothesis can explain chronic OIPN, but also in this case is not the only possible explanation for the occurrence of this side effect. Neuroinflammatory events, either related to or independent from oxidative stress, might be triggered by OHP administration [26,27], and recently, topiramate, a well-known voltage-gated Na+ channel modulator devoid of antioxidant properties, was reported to be able to modify acute as well as chronic OIPN [15]. Finally, the DRG neurons selective expression of plasma membrane transporters’ (e.g., copper transporters such as CTR1, organic cation transporters such as OCT1 and 2) ability to bind and transport platinum compounds has been advocated as a major determinant in the selective toxicity of OHP toward DRG neurons [5,28,29].

Another unsettled issue is represented by the marked difference in the severity of OIPN in different individuals with the same initial conditions and treated with the same OHP schedule. Animal evidence of different susceptibility in different mice strains suggested a genetic background to explain this difference [4], but the several studies performed in OHP-treated patients failed so far to provide firm confirmation [30,31].

## 2. Prevention and Treatment of OHP-Related Peripheral Neurotoxicity

Understanding the mechanism(s) at the basis of OIPN would be critical in the attempt to identify druggable targets and to apply a rationale-based approach to the neuroprotection clinical trials. Unfortunately, on the background previously described, this is not yet entirely feasible and several of the pharmacological approaches fail to have a sound rationale. Non-pharmacological trials are in most cases designed to achieve symptomatic relief, rather than to prevent OIPN. As a matter of fact, the most recent systematic reviews approaching the treatment of OIPN did not find any conclusive evidence of efficacy for any of the tested treatments [32,33].

Several preclinical in vitro and in vivo studies are available, but currently it is difficult to extract from these studies ready-to-use information. Therefore, in order to provide unbiased information potentially closer to application in clinical practice to provide relief from OIPN in adults, we extracted from the public database ClinicalTrial.gov provided by the U.S. National Library of Medicine (https://clinicaltrials.gov/ct2/home, accessed on 1 April 2020) all the registered trials in OHP-treated patients in their starting phase, recruitment phase or completed but still registered, with the last update posted later than January 2016. These studies will be used to identify some of the future direction of prevention/treatment studies in OIPN, and also to discuss methodological issues that might limit the interpretation of the results and their translation into clinical practice.

From this search 11 pharmacological (Table 1) and 5 non-pharmacological (Table 2) trials were identified. They will be revised with particular attention paid to the hypothesis leading to the study, to the trial strengths and weaknesses, and to the outcome measures proposed to test the efficacy of the neuroprotective approach.

### 2.1. Pharmacological Studies

#### 2.1.1. Riluzole

In OHP-treated animals, riluzole prevents the excessive accumulation of glutamate [34], and recently it was evidenced the benefit of riluzole for sensorimotor and painful disorders of the peripheral nervous system, preventing OHP-induced peripheral nerve functional and morphological alterations, as well as related OHP-induced comorbidities [35]. It has been postulated that riluzole could exert its neuroprotective action through interaction with potassium channels of the K2P family (TREK, TRAAK) [36,37].

This is a phase II, randomized, placebo-controlled, parallel, double-blind, multicenter, prevention trial. Patients enrolled are adults with a diagnosis of stage II/III colorectal cancer to be treated with a simplified FOLFOX4 regimen in the adjuvant setting. Riluzole at a dose of 50 mg twice a day will be administered during each chemotherapy cycle, beginning 7 days before the beginning of chemotherapy and ending 2 weeks after the start of the last cycle of chemotherapy. Administration and duration of OHP treatment are equivalent in the control and riluzole groups. The primary endpoint of the study is the Quality of Life Questionnaire-Chemotherapy-Induced Peripheral Neuropathy (QLQ-CIPN20) released by the European Organization for Research and Treatment of Cancer (EORTC) assessed 3 months after initiation of OHP- based chemotherapy (1 cycle = 14 days).

#### 2.1.2. L-Carnosine

This open-label prevention study evaluated the prophylactic effect of exogenous L-Carnosine to prevent oxidative stress. Patients received L-Carnosine (500 mg per day, orally, PO) together with their chemotherapy. Assessment of peripheral neuropathy was done primarily using neuropathy grading score of the National Cancer Institute-Common Toxicity Criteria for Adverse Events (NCI-CTCAE, version 4.0) and additionally also oxidative stress markers were measured by ELISA (Nrf2 induced oxidative stress pathways (GSH), NF-KB anti-inflammatory pathway (TNF-alpha), pro-apoptoic signals (caspase 3).

L-carnosine, formulated in combination with alpha-lipoic acid, was tested in an animal model of OIPN showing a significant reduction in neuropathic pain and good tolerability. It was suggested that the calcium-binding carnosine moiety is able to exert a persistent activity through a synergically stabilized binding to TRPA1 by covalent binding to the channel through the lipoic acid residue [38].

#### 2.1.3. Lidocaine

Although local application of lidocaine has been used in patients with refractory neuropathic pain, the possibility to use the intravenous (IV) way of delivery to treat patients with chemotherapy-induced neuropathic pain was reported. In that small study (9 patients, only 3 of them treated with OHP) the IV administration of lidocaine (1.5 mg/kg in 10 min followed by 1.5 mg/kg/h over 5 h) had direct analgesic effect in CIPN with a moderate long-term effect and seemed to influence the area of cold and pinprick perception [39].

This is a pilot study divided in two subsequent phases aimed at determining if IV lidocaine is a tolerated and effective treatment to reduce the severity of OHP-induced cold hypersensitivity in patients treated with modified FOLFOX6 (mFOLFOX6) chemotherapy. The efficacy of the treatment is assessed using as primary outcome measure the intensity of OHP-induced cold hypersensitivity after 12 weeks of treatment. The study design indicates a first part of the trial as an open-label trial allowing to determine the optimal and tolerable dose regimen of IV to be subsequently tested in a randomized, double-blind, placebo-controlled trial in patients with advanced colorectal cancer receiving OHP chemotherapy

#### 2.1.4. Venlafaxine

In this pilot randomized, placebo-controlled, double blind study on 50 patients, venlafaxine extended release (37.5 mg) or placebo, twice daily, was delivered through their last dose of OHP (FOLFOX) and then titrated off. The results of the study using a primary endpoint based on the EORTC CIPN20 questionnaire have been reported and their analysis failed to demonstrate any efficacy of venlafaxine [40].

#### 2.1.5. Calmangafodipir

Calmangafodipir (PledOx^®^), a compound derived from the contrast agent used in magnetic resonance imaging mangafodipir, possessing antioxidant and MnSOD-mimetic activities [41,42], was evaluated in a clinical trial of 173 patients with colorectal cancer treated with OHP-based chemotherapy and in this phase II randomized, placebo-controlled, double blind-trial, pre-treatment with PledOx significantly reduced the neurotoxicity of the treatment, without reducing its antineoplastic efficacy [43].

PledOx^®^ efficacy is currently is under evaluation in 2 separate phase 3, multicenter, double-blind, placebo-controlled trials. The first trial (POLAR-M) is in patients with metastatic colorectal cancer to be treated with mFOLFOX6 chemotherapy regimen for at least 3 months. Patients are randomized to one of three treatment arms: arm A, PledOx^®^ (2 µmol/kg, IV infusion on the first day of each chemotherapy cycle) + mFOLFOX6 chemotherapy; arm B, PledOx^®^ (5 µmol/kg) + mFOLFOX6 chemotherapy; arm C: Placebo + mFOLFOX6 chemotherapy. The second trial (POLAR-A) is a prevention study of chronic OIPN in patients with stage II/III colorectal cancer to be treated with mFOLFOX6 chemotherapy for up to 6 months in the adjuvant setting. In the POLAR-A trial patients receive PledOx^®^ (5 µmol/kg, IV infusion on the first day of each chemotherapy cycle) + mFOLFOX6 chemotherapy or placebo + mFOLFOX6 chemotherapy. The primary outcome measure in both POLAR-M and POLAR-A is the proportion of patients scoring 3 or 4 in at least 1 of the first 4 items of the Functional Assessment of Cancer Therapy/Gynecologic Oncology Group-Neurotoxicity (FACT/GOG-NTX-13, i.e., FACT/GOG-NTX-4) questionnaire 9 months after the first dose of PledOx or placebo.

On 6 April 2020, PledPharma AB closed its pivotal phase III program POLAR with lead candidate PledOx^®^ after a recommendation from the independent Drug Safety Monitoring Board (DSMB) to stop the studies due to severe allergic reactions observed after repeated dosing. Allergic hypersensitivity reactions are not uncommon during platinum-based chemotherapy, but the DSMB recommendation implies that there might be an increased risk in subjects co-treated with PledOx^®^. A total of 590 patients have been randomized in the POLAR program, of which 420 have completed more than six cycles of treatment and about 250 subjects have completed more than nine cycles. These data will enable efficacy and safety evaluation and an assessment of the benefit/risk of PledOx^®^.

#### 2.1.6. Pregabalin

The rationale behind this study is that pregabalin is a drug that can decrease neuronal hyperexcitability, it is active in relieving neuropathic pain and reaches stable plasma levels after a titration period of only a few days.

In the present Phase III, randomized, double-blind, placebo-controlled study, the hypothesis that pregabalin administrated exclusively for three days before and three days after the OHP infusion is able to prevent the occurrence of pain secondary to both the acute and chronic OIPN was tested. The efficacy of the treatment was primarily based on the presence of OHP-induced painful neuropathy assessed with the Brazilian version of the Douleur Neuropathique 4 Questionnaire (DN4) and on the intensity of pain based on the numeric pain scale (11 points) of the Brief Pain Inventory, six months after treatment discontinuation.

In this prevention study, the results allow the conclusion that the preventive use of pregabalin during OHP infusions was safe, but did not decrease the incidence of chronic pain related to OIPN [44].

#### 2.1.7. Duloxetine

Duloxetine, a serotonin–norepinephrine reuptake inhibitor, is the only recommended treatment for painful chemotherapy-induced peripheral neuropathy [45]. Moreover, an exploratory responder analysis suggests that patients with OHP-induced painful neuropathy are more likely to experience a benefit from duloxetine than patients with paclitaxel-induced neuropathy [46]. However, duloxetine is not completely effective, nor does it works for everyone, thus optimizing the treatment schedule and identifying predictors of duloxetine response is a priority.

This randomized, double-blind, placebo-controlled phase II/III study investigates the best dose of duloxetine and how well it works in preventing pain, tingling, and numbness caused by treatment with OHP in patients with stage II–III colorectal cancer. In phase II of the study patients receive duloxetine 30 mg PO once daily during week 1, duloxetine at the same dose and placebo once daily during weeks 2–16, followed by duloxetine 30 mg PO once daily during week 17, if tolerated; in the second arm, patients receive duloxetine 30 mg PO once daily during week 1, duloxetine at the dose of 60 mg PO once daily during weeks 2–16, followed by duloxetine 30 mg PO once daily during week 17 if tolerated; in the third arm, patients receive only placebo during the entire duration of the study. In phase III of the study, patients receive placebo or the most promising dose of duloxetine from phase II PO once daily in the absence of unacceptable toxicity.

#### 2.1.8. Lorcaserin

The selective 5-HT_2C_ receptor agonist lorcaserin has effects on a range of behaviors and physiological functions. Preclinical studies with lorcaserin initially evidenced its effect on food intake and weight gain, but have now expanded to effects on appetitive aspects of feeding behavior and models of binge-eating. A significant number of studies have also shown that lorcaserin alters behaviors related to drug use and addiction. Finally, potential clinically relevant effects of lorcaserin have also been reported in models of pain and seizure-like activity [47].

The first trial is a randomized phase II study comparing lorcaserin (10 mg twice daily for 180 days) versus duloxetine (60 mg once daily for 180 days) for the treatment of chronic OIPN. Three primary outcome measures are identified: Pain Numerical Rating Scale, Pain Score 0-10 Numerical Rating Scale using the Brief Pain Inventory Short Form, and FACT/GOG-NTX questionnaire, all in a time frame of 180 days. The second study is a phase I open-label trial investigating how well lorcaserin (at a starting dose of 10 mg twice daily) works in treating chemotherapy-induced peripheral neuropathy in patients with stage I-IV gastrointestinal or breast cancer. The primary outcome measure is the improvement in balance evaluated using a measure of postural control calculated as the root-mean-squared amplitude of the center of pressure excursion for the medial-lateral axis of the body.

#### 2.1.9. TRK-750

This crossover randomized study is aimed at investigating the safety, tolerability, pharmacokinetics, and pharmacodynamics of TRK-750 (Toray Industries, Inc, Chuo-ku, Tokyo, Japan in colorectal cancer patients following OHP containing chemotherapy in the adjuvant setting and developing chronic OIPN. The trial’s primary outcome measures are the incidence of adverse events as assessed by NCI-CTCAE version 5.0 (time frame up to week 28), the proportion of patients with clinically significant changes in vital signs (i.e., supine blood pressure, supine pulse rate, respiratory rate, oral body temperature), electrocardiogram, and in clinical laboratory tests.

### 2.2. Non-Pharmacological Studies

#### 2.2.1. Repetitive Transcranial Magnetic Stimulation

Repetitive Transcranial Magnetic Stimulation (rTMS) is a noninvasive form of brain stimulation in which a changing magnetic field is used to provide electric current at a specific area of the brain through electromagnetic induction. When these pulses are administered in rapid succession, they can produce longer-lasting changes in brain activity. rTMS has been shown to be a safe and well-tolerated procedure that can be an effective treatment for patients with depression who have not benefitted from antidepressant medications or cannot tolerate antidepressant medications due to side-effects. However, the use of rTMS in headache and pain, as well as in other psychiatric and neurological conditions has been proposed [48].

In the first study patients with OIPN are randomized to receive high-frequency rTMS or sham rTMS at an intensity set at the lowest stimulator output that can generate similar noise to the real rTMS. The rTMS is delivered over the cortical area M1 (hand representation) of dual hemispheres with 10 trains of 10 Hz pulses for 10 s, with a total of 1000 pulses per hemisphere. The rTMS is delivered as a daily session for five consecutive days, followed by two fortnightly maintenance sessions during the follow-up period after the completion of five daily sessions. The primary outcomes of the study are the effective attitude and perceived effectiveness measured by using the Chinese version of the Patients’ Global Impression of Change (PGIC) scale. The other domains of acceptability are evaluated using a qualitative approach. In the second study (Table 2, #2) patients with OIPN are randomized to be treated as follows: rTMS over 30 min for 10 sessions over 10 business days, sham rTMS over 30 min for 10 sessions over 10 business days, or standard of care. Change in perceptions of chemotherapy-induced peripheral neuropathy from baseline up to 1 month is assessed by the Pain Quality Assessment Scale, a 20-item measure developed to quantify quality and intensity of neuropathic pain derived from the Neuropathic Pain Scale.

#### 2.2.2. Strength and Balance Training Program

The rationale of these studies is to test the hypothesis that lifestyle-related factors can aid in preventing or reducing the neurological side effects of chemotherapy, as such factors may promote self-management options for patients suffering from OIPN. However, the current evidence in support of this hypothesis is weak and many of the studies reporting positive results have serious limitations, including small sample sizes and heterogeneity in trial design and measurements of neurotoxicity [49].

The specific purpose of a small pilot study is to evaluate the effects on strength, balance, and neuropathic symptoms (numbness, tingling, pain, weakness) of a 12 week, bi-weekly, 60-min, group exercise program designed to improve lower extremity strength and balance in persons with OIPN. In view of the very low number of patients, the primary endpoint of the study is to evaluate feasibility, assessed by the calculation of the percentage of patients screened to percentage enrolled in the study and the percentage enrolled to the percentage that completes the study.

#### 2.2.3. Diet

This study is aimed at determining whether a specific nutritional therapy, a polyamine deprived diet, may prevent acute OIPN in patients receiving FOLFOX4. The primary outcome measure of this study focused on the acute symptoms of OIPN is the cold detection threshold assessed on day 42 of treatment. Patients are randomized to receive a polyamines-depleted diet using 2–4 cans per day of Polydol^®^ associated with predefined menus low in polyamines or 1 can per day of Polydol^®^ associated with predefined menus with a normal average in polyamines for 107 days.

#### 2.2.4. Henna Application

Henna is one of the herbal extracts used in the treatment of diabetic cutaneous ulcers, a severe condition associated with small nerve fiber loss [50]. This pathological event is common also in several types of painful neuropathies occurring after the administration of different anticancer drugs, including OHP.

This study includes two groups of women treated with OHP-containing chemotherapy: in the intervention group, henna is applied to the hands and feet after the 2nd and 3rd chemotherapy cycles. The control group undergoes only the routine treatment. Patients in the intervention group are instructed to apply henna before going to bed at night, and after waking up in the morning and to wash with only water. The primary outcome measure of the study is the Chemotherapy-Induced Peripheral Neuropathy Assessment Tool (CIPNAT) filled before the 2nd, 3rd, and 4th chemotherapy cycles.

## 3. Conclusions

Although this is a non-systematic review of the clinical trials focused on prevention or treatment of OIPN, it provides several hints for reflection that deserve to be highlighted because they contribute to the current uncertainty of the best approach to OIPN.

First, several studies do not rely on evidence-based pathogenic rationale, at least if one considers the most solid hypothesis generated from the in vivo preclinical studies. This is true particularly for the treatment studies, but also some of the prevention trials are similarly flawed.

Second, the sample size in most studies is low and it seems unlikely these trials can eventually provide reliable results. This assumption emerges from the comparison between the planned accrual number and the literature evidence based on epidemiologic data in OIPN that suggest the need for larger cohorts in most of these studies.

Third, the assessment methods are highly variable and only a few of them have been specifically designed for OIPN investigation, or at least and more generally, for chemotherapy-induced peripheral neurotoxicity. This variability reflects the absence of a widely accepted gold-standard in OIPN assessment, a major issue in the design and evaluation of similar trials, but it also should point to the need for using only outcome measures already validated in this specific clinical setting.

Fourth, frequently, there is not a clear distinction between acute and chronic OIPN, in terms of recruitment as well as of assessment methods, despite only chronic OIPN is a dose-limiting toxicity.

Finally, the time points of assessment are not adequate in most of the trials aimed at assessing the effect of chronic OIPN. In fact, chronic OIPN typically ensues during treatment, but it peaks only several weeks after the completion of OHP administration (the “coasting” phenomenon), then it can revert or become permanent. For this reason, early time points are not really informative, and long term observations are warranted to really capture the most clinically-relevant aspects of this form of OIPN, i.e., its long-lasting course.

Based on these observations, as well as on the results of the published clinical trials, it must be acknowledged that improvement in clinical trials design is still required [51] and that very careful attention must be paid to the clinical course of OIPN and to the assessment methods [52,53,54] in order to gain useful and reliable information for the possibility of preventing or treating this severe side effect of OHP administration. Among the other main critical issues in clinical trial designs, specific attention should be paid to proper sample size estimation based on solid primary endpoints able to reflect the most relevant clinical problem for each drug. Eligibility criteria represent another critical aspect in terms of translation of the study results into real-life practice, for instance regarding possible co-morbidities such as diabetes or pre-existing neuropathies that are typically excluded from clinical trials but are frequently encountered in clinical practice. Moreover, trials should clearly disclose in their design if they are investigating a preventive or a symptomatic treatment, since drugs with an expected symptomatic effect have been apparently investigated using outcome measures more appropriate for preventive agents, and vice versa. Additionally, prevention trials must take into great consideration and properly address exclusion of any interference with the ongoing cancer treatment. Finally, the identification of reliable biomarkers able to predict the clinical course of OIPN would represent a relevant improvement in the clinical research toolkit. Currently, measurement of the level of neurofilament light chains in plasma or serum samples seems to be the most promising candidate based on preclinical evidence obtained using different neurotoxic chemotherapy regimens [55,56], but further validation in the clinical setting is required.

While this review was focused on ongoing clinical trials based primarily on established pathogenetic hypothesis, a huge amount of additional preclinical studies dealing with OIPN prevention have been recently published, and in most cases they were exploring new and potentially interesting targets. An extensive revision of these preclinical studies goes beyond the scope of this review, but some of these new and original approaches (not yet tested at the clinical level) might be used as examples of innovative research and provide useful information for future trials. For instance, neuroinflammation modulation achieved using anti-macrophage-derived high mobility group box 1 (HMGB1) neutralizing antibodies reduced the severity of OIPN in a mice model [57]. On a similar hypothesis stands the observation that the synthetic derivative of progesterone, 17α-hydroxyprogesterone caproate (HPGC) is able to prevent OHP-induced allodynia as well as glial activation in mice [58]. New methodological approaches might also be helpful to drive the preclinical investigation. As an example, screening several sets of small-molecule chemical libraries in silico and subsequently validating the results with an in vitro high-throughput phenotypic assay, fulvestrant, a clinically approved drug for the treatment of breast cancer in postmenopausal women, was identified as a potential neuroprotective agent, and this hypothesis was successfully tested in a rat model [59]. Finally, investigations might also originate from the experience gained in other neurological diseases. For instance, the multiple sclerosis oral treatment dimethyl fumarate acting via up-regulation of the nuclear factor-erythroid-2-related factor 2 (Nrf2)-dependent antioxidant response was tested in vitro, showing positive results against oxaliplatin-, cisplatin-, and bortezomib- (but not paclitaxel-) induced inhibition of neurite outgrowth [58]. A different hypothesis is at the basis of the use of the anti-Parkinson’s disease benztropine, an anti-histamine and dopamine re-uptake inhibitor with neuroprotective effects in neurodegenerative neurological disease models as well as in experimental diabetic neuropathy, that reduced OIPN severity in OHP-treated mice synergizing its anti-tumoral effect [60].

Another important aspect highlighted by preclinical studies is the involvement of the central nervous system in OIPN [21,22]. However, most of the studies regarding the central effects of neurotoxic chemotherapy are only focused on cognitive impairment (“chemofog” or “chemobrain”) [61,62,63], while very little attention has been paid so far to this important aspect, particularly regarding the painful component of OIPN. It is, therefore, advisable that this lack of information will be addressed by well-conducted clinical studies profiting from the possibility to investigate the central nervous system at the functional level with magnetic resonance imaging or to modulate its activity using transcranial stimulation because this approach might open the way to new therapeutic attempts [64].

So far, the most effective strategy to limit the severity of chronic OIPN still remains schedule modification. Under this perspective, the results of the SCOT study, a large (more than 6000 patients) international, randomized, non-inferiority trial in patients with high-risk stage II and stage III colorectal cancer receiving OHP-containing treatment regimens (CAPOX, FOLFOX), evidenced that grade 2 or worse OIPN was more common in the group treated for 6 months (58% of patients with safety data) than in the 3-month group (25%). Remarkably, 3 months of OHP-containing adjuvant chemotherapy was non-inferior to 6 months of the same therapy, thus suggesting this approach to effectively reduce incidence of severe OIPN [65].

It can be concluded that the prevention and treatment of OIPN still remains an important and unmet clinical need, and that further, high-quality research is mandatory in order to achieve reliable and effective results.

## Figures and Tables

**Table 1 cancers-12-01370-t001:** Pharmacological studies including oxaliplatin-treated patients registered at ClinicalTrial.gov on 1 April 2020.

#	Title of the Trial	Number of Patients	Last Updated	Allocation	Intervention Model	Masking	Status
1	Effectiveness Assessment of Riluzole in the Prevention of Oxaliplatin-induced Peripheral Neuropathy	210	Oct 9, 2019	randomized	parallel assignment	quadruple (Participant, Care Provider, Investigator, Outcomes Assessor)	Not yet recruiting
2	L-carnosine Prophylactic Effect on Oxaliplatin Induced Peripheral Neuropathy in GIT Cancer Patients	65	Apr 24, 2017	randomized	parallel assignment	none	Completed
3	Lidocaine for Oxaliplatin-induced Neuropathy	38	Jan 22, 2020	randomized	parallel assignment	quadruple (Participant, Care Provider, Investigator, Outcomes Assessor)	Recruiting
4	Venlafaxine in Preventing Chronic Oxaliplatin-Induced Neuropathy In Patients Receiving Combination Chemotherapy	50	Sep 26, 2019	randomized	parallel assignment	double (Participant, Investigator)	Completed
5	Preventive Treatment of Oxaliplatin Induced Peripheral Neuropathy in Metastatic Colorectal Cancer (POLAR-M)	420	Mar 27, 2020	randomized	parallel assignment	triple (Participant, Care Provider, Investigator)	Recruiting
6	Preventive Treatment of OxaLiplatin Induced peripherAl neuRopathy in Adjuvant Colorectal Cancer	301	Feb 20, 2020	randomized	parallel assignment	triple (Participant, Care Provider, Investigator)	Active, not recruiting
7	Evaluate The Efficacy and Safety Of Pregabalin In Prevention, Reduction of Oxaliplatin-Induced Painful Neuropathy	200	May 9, 2017	randomized	parallel assignment	quadruple (Participant, Care Provider, Investigator, Outcomes Assessor)	Completed
8	Duloxetine to Prevent Oxaliplatin-Induced Peripheral Neuropathy in Patients With Stage II–III Colorectal Cancer	327	Feb 20, 2020	randomized	Sequential Assignment	double (Participant, Investigator)	Not yet recruiting
9	Comparing Lorcaserin Versus Duloxetine for the Treatment of Chemotherapy-Induced Peripheral Neuropathy	50	Apr 4, 2019	randomized	parallel assignment	coded bottles	Not yet recruiting
10	Lorcaserin in Treating Chemotherapy-Induced Peripheral Neuropathy in Patients With Stage I–IV Gastrointestinal or Breast Cancer	30	Dec 19, 2019		Single Group Assignment	none	Not yet recruiting
11	A Study to Investigate the Safety and Efficacy of TRK-750 for the Treatment of Patients With CIPN (Chopin Study)	240	Feb 25, 2020	randomized	Crossover Assignment	triple (Participant, Care Provider, Investigator)	Not yet recruiting

**Table 2 cancers-12-01370-t002:** Non-pharmacological studies including oxaliplatin-treated patients registered at ClinicalTrial.gov on 1 April 2020.

#	Title of Trial	Number of Patients	Last Updated	Allocation	Intervention Model	Masking	Status
1	Use of Repetitive Transcranial Magnetic Stimulation in Cancer Patients With Oxaliplatin-Induced Peripheral Neuropathy	60	Sep 27, 2019	randomized	parallel assignment	Double (Participant, Outcomes Assessor)	Not yet recruiting
3	Rtms in Improving Neuropathy in Patients With Stage I–IV Cancer Who Have Received Oxaliplatin Chemotherapy	60	Mar 6, 2020	randomized	parallel assignment	Double (Participant, Outcomes Assessor)	Recruiting
2	Pilot Study of Strength and Balance Training Program for Persons With Oxaliplatin Induced Neuropathy	4	Dec 29, 2016		single group	none	Completed
4	Prevention of Oxaliplatin-induced Neuropathic Pain by a Specific Diet	80	Jul 11, 2017	randomized	parallel assignment	single (Participant)	Completed
11	The Preliminary Effects of Henna on CIPN	60	Dec 17, 2019	randomized	parallel assignment	single (Outcome Assessor)	Completed

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
