# Peer review of "Management of Oxaliplatin-Induced Peripheral Sensory Neuropathy"

_cancers, 2020, doi:10.3390/cancers12061370_

Round 1

Reviewer 1 Report

This review article accounts the details of ongoing clinical trials associated with the management of oxaliplatin-induced peripheral sensory neuropathy. This concise review article highlights some key problems associated with ongoing clinical trials associated with oxaliplatin-induced peripheral sensory neuropathy. However, this review is unable to provide clear possible solutions to address the problems associated with these clinical trials. Further, some topic could also be included and elaborated. For example, comparative analysis of clinical trials of other platinum-based drugs with oxaliplatin, promising candidates against oxaliplatin-induced peripheral sensory neuropathy for clinical trials (based on animal experiments) etc.

Author Response

First of all, thank you for your very valuable comments. Below our replies.

C: This review article accounts the details of ongoing clinical trials associated with the management of oxaliplatin-induced peripheral sensory neuropathy. This concise review article highlights some key problems associated with ongoing clinical trials associated with oxaliplatin-induced peripheral sensory neuropathy. However, this review is unable to provide clear possible solutions to address the problems associated with these clinical trials.

A: It is correct that in our previous version we only mentioned and very rapidly address this aspect. In this revised version the most relevant issues and their possible solutions have been discussed in more detail in the Conclusion section (lines 348-359)

C: Further, some topic could also be included and elaborated. For example, comparative analysis of clinical trials of other platinum-based drugs with oxaliplatin, promising candidates against oxaliplatin-induced peripheral sensory neuropathy for clinical trials (based on animal experiments) etc.

A: This is a very good point, but at the same moment it would require (to really be fully addressed) a huge extension of the review that, unfortunately, would lead to the same conclusion since there is no better evidence for neuroprotection agents using other platinum-based drugs; moreover, given the much more widespread use of oxaliplatin compared to cisplatin or carboplatin (other platinum-based drugs have only a regional or experimental use) the vast majority of the recent studies are focused on it. However, we agree that more relevance to the preclinical studies might be beneficial for the review and we  have now added (lines 363-386) a new paragraph where several very recent preclinical studies used as examples of innovative investigational attempts

Reviewer 2 Report

This is an interesting update about the clinical trials in oxaliplatin-treated patients. Unfortunately no relevant news appear, so it is of pivotal importance that experts in the field, as these authors are, continue to analyze tdoi: 10.1016/j.neuropharm.2017.12.020he problem and highlight difficulties.

Only minor points could be improved. Interestingly, the authors point out the term "neuroprotection" evaluating the "the outcome measures proposed to test the efficacy of the neuroprotective approach....". For this purpose a clear differentiation of the approaches in really neuroprotective (I mean preventive neurotoxicity or neurorestorative) or only symptomatic could be useful, e.g. we cannot aspect protection by lidocaine but yes by carnosine; the point is debated for pregabalin or venlafaxine, what about riluzole or other? 

Again, did these studies evaluate biomarkers of neurotoxicity? e.g. neurofilament (as the same authors recently described) or other recently shown like VEGF-A (doi: 10.1016/j.neuropharm.2017.12.020). Probably not, the highlight of the relevance of these kind of measurements, and the encouragement to evaluate biomarkers during clinical trials, should be very necessary.

Finally, introducing the problem the authors mentioned the role of the central nervous system toxicity of this "peripheral" neuropathy. The relevance of the central damage and response is strongly increasing, so the suggestion is to expand this point reporting CNS alterations and possible new targets (PPARy doi: 10.3389/fnins.2019.00907). It could be also commented that a not enough attention to the CNS is one of the reasons of an unsatisfactory therapy

Author Response

First of all, thank you for your valuable comments. Below our replies.

This is an interesting update about the clinical trials in oxaliplatin-treated patients. Unfortunately no relevant news appear, so it is of pivotal importance that experts in the field, as these authors are, continue to analyze the problem and highlight difficulties.

Only minor points could be improved.

C: Interestingly, the authors point out the term "neuroprotection" evaluating the "the outcome measures proposed to test the efficacy of the neuroprotective approach....". For this purpose a clear differentiation of the approaches in really neuroprotective (I mean preventive neurotoxicity or neurorestorative) or only symptomatic could be useful, e.g. we cannot aspect protection by lidocaine but yes by carnosine; the point is debated for pregabalin or venlafaxine, what about riluzole or other? 

A: This is an excellent point that now has been highlighted in the Conclusion section (lines 353-356), in the new paragraph discussing improvements in the design of new clinical trials in OIPN. In fact, sometimes the results of  "preventive" trials are evaluated with outcome measures that would be more appropriate for symptomatic drugs, and vice versa

C: Again, did these studies evaluate biomarkers of neurotoxicity? e.g. neurofilament (as the same authors recently described) or other recently shown like VEGF-A (doi: 10.1016/j.neuropharm.2017.12.020). Probably not, the highlight of the relevance of these kind of measurements, and the encouragement to evaluate biomarkers during clinical trials, should be very necessary.

A: Also this is a very relevant aspect of the clinical studies designs and we have added a comment highlighting its importance in the Conclusion section (lines 358-362)

C: Finally, introducing the problem the authors mentioned the role of the central nervous system toxicity of this "peripheral" neuropathy. The relevance of the central damage and response is strongly increasing, so the suggestion is to expand this point reporting CNS alterations and possible new targets (PPARy doi: 10.3389/fnins.2019.00907). It could be also commented that a not enough attention to the CNS is one of the reasons of an unsatisfactory therapy

A: Although the target of this review is the peripheral neurotoxicity of oxaliplatin, this comment raises an important and under-investigated aspect that has now been highlighted in the Conclusion section (387-395)